# Immersive virtual reality simulation for undergraduate nursing students: Enhancing mental health care for migrants - A mixed method study protocol

Geneveave Barbo[1]*, Donald Leidl[2], Marjorie Montreuil[3], Hua Li[4], Solina Richter[4], Pammla Petrucka[1]

1 College of Nursing, University of Saskatchewan, Regina, Saskatchewan, Canada, 2 Faculty of Nursing, University of New Brunswick, Moncton, New Brunswick, Canada, 3 Ingram School of Nursing, McGill University, Montreal, Quebec, Canada, 4 College of Nursing, University of Saskatchewan, Saskatoon, Saskatchewan, Canada

* g.barbo@usask.ca

## Abstract

The evolving field of nursing education increasingly integrates innovative methods such as immersive virtual reality (IVR) to improve training outcomes. This protocol paper outlines a study that addresses a significant gap by using IVR to improve mental health care training for undergraduate nursing students, focusing particularly on migrants who frequently encounter access barriers such as stigma, discrimination, and cultural differences. Traditional training methods frequently fail to provide the experiential learning necessary for nursing students to develop deep empathy, cultural competence, cultural humility, and advanced communication skills. A multiphase, sequential explanatory mixed methods design will be employed in this study, which encompasses three phases: development of IVR simulation, a one-group pre- and post-quasi-experimental design, and an interpretive description approach. Participants will include undergraduate nursing students from the University of Saskatchewan and McGill University. In Phase 1, an integrative review will establish the foundation for the simulation, the findings of which will inform the design of initial simulation drafts on the Unity platform. These drafts will be reviewed by an advisory committee, consisting of migrants experiencing mental health challenges, nursing students, educators, and nurses specialized in migrant health care. Feedback from the committee will be integrated before progressing to Phase 2. Phase 2 involves collecting data through pre- and post-intervention questionnaires completed by participants. This data will be analyzed using descriptive and inferential statistics to assess the impact of the IVR simulation and to inform the next phase of the study. In Phase 3, participants will engage in semi-structured interviews. This phase will employ concurrent data collection and analysis along with constant comparative analysis in an iterative process. Following separate analyses of quantitative and qualitative

**Data availability statement:** No datasets were generated or analysed during the current study. All relevant data from this study will be made available upon study completion.

**Funding:** The author(s) received no specific funding for this work.

**Competing interests:** The authors have declared that no competing interests exist.

data, the results will be synthesized to provide a comprehensive interpretation of the findings. The expected outcomes include greater acceptance of IVR as a training tool, positive shifts in student attitudes and behaviours towards migrants with mental health difficulties and enhanced cultural competence. This innovative approach could standardize the use of IVR in nursing curricula, making it a fundamental component of nursing education aimed at preparing students for a diverse and inclusive healthcare environment.

## Introduction

In recent years, nursing education has been evolving to incorporate innovative teaching methods aimed at enhancing the skills and competencies of nursing students [1,2]. One such innovation is the use of immersive virtual reality (IVR) simulation, which has shown promise in various educational settings [3–6]. However, there remains a gap in applying this technology specifically to train nursing students in providing mental health care for migrants—a population that faces unique and complex challenges.

Migrants often experience significant mental health stressors related to pre-migration, migration, and post-migration phases [7]. These stressors can include stigma, discrimination, cultural differences, and language barriers; all of which contribute to significant challenges in accessing appropriate mental healthcare [8–13]. Nursing students, therefore, need to be well-prepared to recognize and address these challenges effectively. Traditional classroom-based education and clinical rotations, while valuable, may not fully equip students with the experiential learning necessary to develop deep empathy and cultural competence [14,15]. This protocol paper therefore details the methodology of a research study aiming to address this gap.

## Background

"Extended reality" has recently begun being used as an umbrella term encompassing augmented reality, mixed reality, and virtual reality (VR) [16]. The term "virtual reality" was coined by Jaron Lanier, a computer scientist and founder of the Virtual Programming Laboratory in 1987 [4,17]. Initially, VR represented various hardware and software technologies associated with providing users with elements of a virtual environment, such as Sensorama Simulator, Second Life, and Cave Automatic Virtual Environment [18]. It was only in 2013 that VR technology took off, with the development of affordable first-version head-mounted devices (HMDs) from Oculus Rift and other competitors [18]. According to Hodgson et al. [19], VR hardware reduced its price from 45000 USD in 2006–1300 USD in 2014, making VR use as a medium for delivery of VR software and applications more widespread. Besides the costs, the quality of the VR experience also significantly improved after 2013, a trend that has been continuous to date, with PlayStation and Oculus Rift recently releasing their second version HMDs [18].

Along with the boost of VR technology and applications, related language and terminology are still evolving. VR has been defined as a computer-generated

three-dimensional (3D) environment that simulates real-life experiences and enables users to actively interact with the elements within such environment using external devices (e.g., mouse or joystick) [3,20–23]. VR models the real world by imitating sensory inputs (i.e., auditory, visual, tactile, and olfactory) with digital inputs [3].

Despite no definite consensus about the definition of VR, many authors agree on the types of VR (i.e., non-immersive, semi-immersive, and immersive VR). These categories are based on the degree of immersion presented to the user. Immersion is a technological aspect that enhances an individual's sense of presence while viewing an image [3,18]. By increasing the perceptions of presence of a user, they may psychologically experience the virtual world despite being physically in the real world [22]. As such, the term 'presence' has been commonly referred to as the feeling of "being there" or being in the virtual medium [20]. The illusion is perceptual, but not cognitive, as the perceptual system identifies the events and objects and the brain-body system automatically reacts to the changes in the virtual environment, while the cognitive system slowly responds with a conclusion of what the person experiences is an illusion [24].

Multiple factors influence both immersion and presence. Since presence is associated with one's ability to perceive, varying degrees of presence can therefore be felt by different individuals despite being exposed to the same experience [25]. In addition to individual factors, immersion and presence are also influenced by the quality of the images presented, degree of immersion offered, amount of control and interactivity of the system, quality of hardware, awareness of being observed, and the body's positioning (i.e., standing up versus sitting down) [18,25,26].

Consequently, software that provides little to no immersion, such as in the case of non-immersive VR (e.g., Second Life and Minecraft), offers little to no degree of presence [22]. Although non-immersive VR programs are affordable, since they do not require HMDs and can be easily run on computer screens and mobiles, they fall short of matching the learning benefits associated with IVR [18]. Unlike semi-immersive VR, AR, or MR, which allow users to experience both virtual and real world at the same time by overlapping digital images on top of the real-world environment, IVR completely engages the user in the virtual environment by eliminating any real-world sensory input [22,25]. In so doing, the user becomes entirely immersed in a controlled, virtual environment that simultaneously tracks the users' movements and positions, allowing for a richer and more interactive experience [17,22].

VR offers a ground-breaking and innovative tool for connecting theoretical knowledge with real-world clinical practice for nursing students and professionals [1,2]. Specifically, some positive outcomes related to the application of VR in nursing education include heightened student engagement, visualization of intricate models or situations, better decision-making skills, and enhanced practice of clinical skills as well as improvement in cognitive performance, learning, clinical, and psychomotor skills [3–6]. Despite these benefits, certain drawbacks to VR were also found; they include technical problems, limited avatar choices, VR simulation not adequately realistic for students, difficulties for left-handed users, prescription glasses not fitting comfortably with VR headsets, and potential cybersickness [3,5]. Nevertheless, VR, particularly IVR, has been shown to be appropriate and applicable to all nursing students, regardless of their age, gender, or level of expertise [3].

The application of VR in mental health nursing education, particularly for addressing the needs of migrant populations, is still an emerging field. While multiple studies have examined non-immersive VR in mental health nursing education [27–31], and one study has evaluated IVR [6], no research to date has focused specifically on IVR simulation for migrants with mental health difficulties and access barriers. By simulating the experiences of migrants accessing mental health services, VR can provide nursing students with a deeper understanding of the cultural, social, and psychological factors that impact migrant health. This immersive approach not only enhances clinical and communication skills but also fosters empathy, cultural competence, and cultural humility, which are all crucial for effective nursing practice.

## Study objectives

This study aims to bridge the gap in nursing education by developing and evaluating IVR simulation focused on mental health care for migrants. The specific objectives are:

- Aim 1: To evaluate and explore the acceptability of IVR simulation among undergraduate nursing students for mental health training focused on migrants.

- Aim 2: To measure and investigate the initial effects of IVR simulation on the attitudes of undergraduate nursing students toward migrants with mental health difficulties.

- Aim 3: To assess the initial effects of IVR simulation on the cultural competence of undergraduate nursing students towards migrants.

- Aim 4: To compare the acceptability and initial effects of IVR simulation between nursing students at the University of Saskatchewan and McGill University.

By addressing these objectives, the study seeks to demonstrate the potential of VR as a transformative tool in nursing education, particularly in preparing nursing students to meet the mental health needs of diverse and vulnerable populations.

## Materials and methods

### Research design

This study will follow a multi-phase sequential explanatory mixed methods design, with each phase progressively informing and shaping the subsequent phase of the study. Phase 1 will revolve around the development of the intervention through the completion of an integrative review and participatory research approach. Phase 2, the quantitative portion of the study, will involve a one-group pre- and post-quasi-experimental design. Finally, Phase 3 will follow an interpretive description approach.

### Participants and setting

Registered undergraduate nursing students from the University of Saskatchewan and McGill University will be invited to participate in the study. Those interested in participating will be screened for eligibility to identify those with history of adverse reactions to virtual reality hardware device. Individuals who are deemed as high risk will be excluded from the study.

All cohort years will be included to allow participation from students at different stages of their program. Participants will not be required to have a specific background related to mental health training, as this was not an inclusion criterion. Racial, cultural, and academic diversity will not be controlled for in the recruitment process; instead, these characteristics will be captured through demographic information questions for descriptive purposes, as examining these factors was not a specific focus of the study.

It is also important to note that the curricula differ between the Ingram School of Nursing undergraduate program at McGill University and the College of Nursing at the University of Saskatchewan. This variation will be acknowledged when interpreting results, recognizing that differences in curricular design and content may influence student experiences with the intervention.

### Recruitment and sampling

Recruitment posters will be shared to the program coordinators and/or administrative support staff of both the University of Saskatchewan and McGill University who will disseminate it via e-mail to all undergraduate nursing students enrolled in the programs. More targeted recruitment methods, such as sharing the recruitment posters to specific courses, student associations, or students, will also be performed as needed to reach the desired sample size and ensure that the participant groups from the University of Saskatchewan and McGill University are comparable in terms of size, age, sex,

ethnic background, and year of study in the program. We will aim to recruit a diverse group of participants by age, sex, ethnic background, and years of study in the program. However, to be included in the study, students must be fluent in English and have not yet graduated from their nursing program. Moreover, since the purpose of the study is to determine the acceptability and initial effects of the VR simulations rather than obtaining statistically significant results, convenience, purposive, and snowball sampling will be used to obtain participants.

To determine the appropriate sample size, G*Power, a widely recognized tool for statistical power analysis, was utilized. However, the appropriate effect size, alpha error probability, and desired power had to be determined. For the effect size, since the proposed study is novel, no studies have thus far been performed with similar intervention (i.e., IVR simulations using avatars focused on migrants with mental health difficulties), so closely related studies were be used as reference. Emphasis was placed on studies and systematic reviews that provided the effect size or adequate information in order to calculate the effect size. These studies include Chen et al. [32], Efendi et al. [33], Foronda et al. [5], Lee et al. [6], and Piot et al. [30]. Across these studies the effect size varied from small to large. Hence, a medium effect size (Cohen's $d = 0.5$) was chosen as the most conservative approach based on the studies in the field. Similarly, the alpha error probability was set at the conventional 0.05, with a desired power of 0.80. Additionally, difference between two dependent means was selected for the statistical test given that the same participant will complete the pre- and post-questionnaires. G*Power's computation suggested a sample size of 27 participants is needed to achieve the desired statistical power for a medium effect size. Additionally, considering a low dropout rate of 10%, the calculated sample size was adjusted upwards to account for potential participant attrition, leading to the required sample size of 30 participants. It is important to note that this sample size pertains to only one research site. To conduct a comprehensive comparison analysis between the two sites, University of Saskatchewan and McGill University, each site must recruit 30 participants, that is 60 participants combined. This approach ensures that the comparative analysis across different educational and cultural contexts is adequately powered, providing reliable and generalizable findings from each site.

## Ethical considerations

Research ethics approval will be obtained from both the University of Saskatchewan and McGill University. Prior to the start of data collection in Phase 2, the consent form will be reviewed in detail with all potential participants to ensure informed consent. Consent will primarily be obtained through written signatures on the consent form. In cases where verbal consent is provided, the lead author will sign the participant's name on the form in their presence, witnessing the process.

At the University of Saskatchewan, the lead author was a student and not a faculty member, ensuring no instructional or evaluative power over potential participants. At McGill University, at the start of Phase 2, the lead author held an academic associate position responsible for clinical placements but was not teaching any nursing students directly. Recruitment at both sites was therefore designed to avoid any risk of indirect coercion arising from a faculty–student power dynamic.

To maintain confidentiality, each participant will be assigned an alphanumeric code, which will be used to label all associated materials, including completed questionnaires, audio recordings, video recordings, and transcripts. All identifying information will be removed from the study files, with the exception of the consent forms. These forms will be stored separately from other study materials and securely kept in the principal investigator's OneDrive account. Sharing of data will be considered on a case-by-case basis upon request to protect participants privacy, particularly for the qualitative data collected. Any amendments to the study protocol will be submitted to the appropriate Research Ethics Boards at both the University of Saskatchewan and McGill University for approval.

## Procedure of the study

**Phase 1 – Development of the intervention.** An integrative review will be completed which focuses on aggregating and analyzing literature on how healthcare providers can adequately care for migrants with mental health difficulties who were facing barriers to access, specifically stigma, discrimination, and cultural differences. The findings from this

review will directly inform the conceptual framework and content of the IVR simulation in several important ways. First, case studies identified in the literature will be adapted to create realistic patient backstories and clinical scenarios that reflect authentic challenges in practice. Second, the review will help identify the key skills, approaches, and frameworks necessary for providing competent care to migrants facing mental health challenges. These elements will form the basis of the simulation's core learning objectives and each scene's sub-objectives and tasks.

The review will also inform how these key skills, approaches, and frameworks are transformed into practical communication strategies that participants can apply in the simulation and incorporate directly in their nursing care. These strategies will be integrated as dialogue options for participants who will be assuming the first-person perspective of the nurse caring for the virtual patient, with each option highlighting a particular skill, approach, or framework and guiding the participant along a distinct narrative path. Additionally, the simulation will include at least one dialogue option demonstrating a negative or incorrect response to provide clear contrast and support deeper understanding of best practices. To support the design and management of these dialogue options and multiple branching narrative paths, decision trees will also be developed. Each node will correspond to a dialogue option tied to a specific skill or approach, with narrative consequences and feedback tailored to reinforce learning objectives.

For example, consider the scenario in which the virtual patient, after being triaged, spent five hours in the emergency waiting room and begins to show signs of agitation. The patient paces rapidly and shouts at the triage nurse, "I need help, not just sitting away from everyone! In my country, you go to the hospital, and you get seen right away! It's supposed to be better here!" The participant will be tasked with intervening and will be presented with four dialogue options: one focused on cultural competency, another on cultural humility, a third on building a therapeutic alliance, and a fourth demonstrating a negative response.

The culturally competent response option will involve the nurse saying, "I understand that waiting can be very hard, especially when you're feeling unwell. In your culture, immediate care might be expected, and I'm truly sorry for any distress the delay is causing you. We're working to see you as soon as possible because your well-being is important to us." The cultural humility approach option will feature the nurse stating, "Thank you for sharing your feelings. I recognize that each person's experience and needs are unique, and I may not fully understand everything you're going through. Please help me understand what would make you feel more comfortable right now while we wait. What needs to happen to help you feel more under control now?" For the therapeutic alliance building approach, the nurse will say, "I see this is really tough on you, and I want to be here for you during this wait. It's important to me that we use this time to ensure you feel supported. Let's take some deep breaths together." The dismissive tone, provided for contrast, will include the nurse saying, "Everyone here is waiting, and we're all busy. You need to be patient. We'll get to you when it's your turn, and complaining won't make it any faster."

When the foundational framework is established, the lead author will then build the simulation in Unity with the assistance of ChatGPT-4o and relevant websites, such as Sketchfab.com (for 3D models), Mixamo.com (character rigging and animations), and ElevenLabs.io (voice generation).

Upon the completion of the initial IVR simulation, it will be shared with the members of the advisory committee which will consist of a total of eight members, with equal representation from the two cities (Montreal and Saskatoon). The committee will include: two nurses specializing in mental health care for migrants; two adult migrants facing mental health challenges; four undergraduate nursing students; and two mental health nursing educators. We will also attempt to recruit diverse members (i.e., different ages, sexes, and ethnic backgrounds). Committee members will be recruited from both the University of Saskatchewan and McGill University, as well as from the co-authors' partners and professional networks.

Consultation meetings will be held in groups or individually with the committee members, in-person or via Zoom. These meetings will follow a discussion-based, informal format in which committee members will provide feedback while actively experiencing the simulation. The simulation will be paused at intervals to allow for feedback, note taking, clarifications, and discussion of alternative options before resuming. In cases when committee members are unable to attend

consultation meetings, offline or email correspondence will also be offered as an alternative way to provide feedback, enabling them to review the decision trees and foundational framework or blueprint of the simulation at their convenience.

Feedback from each committee member will be documented or transcribed verbatim as appropriate. Although this feedback will not be formally coded or thematically analyzed in a qualitative analysis framework, it will be systematically reviewed by the research team. The advisory committee's feedback will directly inform revisions to the simulation, specifically improving its interface, usability, realism, dialogue scripts, and cultural representativeness. For instance, committee members will be invited to suggest simplifying dialogue language, improving knowledge translation, and strengthen cultural and patient authenticity. The members of the committee will also be invited to review the study protocol.

Once the feedback of the members of the committee has been gathered and fully integrated, the revised simulation will be compiled and built in Unity as a standalone Windows executable (.exe) application. During the Unity build process, all required assets—including 3D models, textures, audio files, animations, scripts, and configuration data—are automatically packaged into an output directory along with supporting data files and subfolders. These files are consolidated into a dedicated build folder, forming a self-contained application that can be launched independently.

To facilitate easy access for the research team, the compiled simulation will be added to the Steam desktop environment. This process will involve launching Steam, navigating to the user's Library, selecting the "Add a Game" option, and choosing "Add a Non-Steam Game" from the dropdown menu. The executable file will then be located within the Unity build folder and added to the Steam Library. Once added, the simulation will appear as a selectable application within the local Steam Library and can be launched directly in virtual reality mode. The simulation will remain entirely private—it will not be published on the Steam Store. As such, it is only discoverable and accessible from the local device on which it was installed, and cannot be viewed, downloaded, or accessed by other Steam users.

Following installation, the application will test to ensure full compatibility with the headset and controllers, confirming that all tracking, interactions, and scene transitions function as intended. Once the simulation is confirmed to be fully operational and reliably accessible, Phase 2 of the study will begin.

**Phase 2 – One-group pre- and post-quasi-experimental.**  Once a student agrees to participate in the study, each one will be invited to either the University of Saskatchewan or McGill University campus where they will first complete the pre-intervention questionnaires and then view the IVR simulation through a HMD and first-person perspective. Each participant will meet individually with either the lead author or a research assistant in a quiet, private room equipped with a VR-compatible laptop that meets the system requirements for Windows Mixed Reality. This laptop is connected via cable to an HP Reverb G2 headset, which delivers a resolution of 2160×2160 per eye (4320×2160 total) and uses inside-out tracking via integrated camera. While the device is tethered to the laptop, the setup supports for limited physical mobility within a small, open space.

Prior to the participant's arrival, the lead author or research assistant will complete the following hardware setup procedure: powering on the VR-compatible laptop, connecting the headset via its tethered cable, and launching both Windows Mixed Reality and Steam. The head-mounted display and controllers will also be powered on and paired. The IVR simulation software is stored in the Steam personal library and is launched from there when the participant is ready to begin.

The pre-intervention questionnaires consist of demographic information, the Technology Acceptance Model Questionnaire (TAMQ), the Opening Minds Scale for Health Care Providers (OMS-HC) [34,35], and the Nurse Cultural Competence Scale (NCCS) [36]. Demographic information will focus on the respondents' age, biological sex, race-based identity, migration/citizenship status, and current level in the nursing program.

The version of TAMQ that will be used for this study has been developed and tested by Cabero-Almenara et al. [37] based on the original Technology Acceptance Model [38]. TAMQ is a 15-Likert-type item divided into five dimensions: perceived utility (similar to PU construct), PEOU, perception of enjoyment, attitude towards use, and intent to use. Response options vary from 1=Extremely unlikely/disagree to 7=Extremely likely/agree. TAMQ's Cronbach's alpha index represents 0.978, indicating an excellent reliability score [37,39].

 

The OMS-HC is a self-report questionnaire that will be used to measure attitudes and behavioural intentions of healthcare providers towards individuals with mental health conditions [34,35]. OMS-HC has been chosen since it has been developed and tested on Canadian healthcare providers to examine anti-stigma interventions [34]. During the initial psychometric properties testing of its 20-item scale, OMS-HC achieved good internal consistency, Cronbach's alpha = 0.82, satisfactory test-retest reliability, and intraclass correlation = 0.66 (95% CI 0.54 to 0.75), but weak correlation with social desirability component. The scale's items were then reduced to 15 items in Modgill et al. [35], resulting in acceptable internal consistency (α = 0.79) and the ability to detect positive changes in different anti-stigma interventions. OMS-HC has been divided into three factors: (1) attitudes of healthcare providers towards people with mental illness; (2) disclosure/help-seeking; and (3) social distance. It consists of a 5-point Likert scale of strongly disagree, disagree, neither agree nor disagree, agree, strongly agree; each representing a balanced score of 1–5 (i.e., strongly disagree equates to a score of 1 and strongly agree is 5). Each score in every item is added together and the total final score will range from 15 (least stigmatizing) to 75 (most stigmatizing) [35]. Items 2, 6, 7, 8, and 14 require reverse coding [35].

Developed by Perng and Watson [36], the NCCS is a 5-point Likert Scale composed of 20 items, incorporating the following concepts: cultural knowledge, cultural sensitivity, and cultural skill. The emphasis on these concepts was one of the reasons that the NCCS was chosen for this study. NCCS includes choices from strongly disagree (scored as 1) to strongly agree (scored as 5). Total scores therefore range between 20 and 100 with the highest score representing a high cultural competence [36]. According to a systematic review conducted by Osmancevic and colleagues [40], the NCCS "showed moderate quality of evidence for indeterminate structural validity" (p. 12). In contrast, they found a high quality of evidence supporting its sufficient internal consistency, reliability, and construct validity [40]. Additionally, the Cronbach's alpha reliability coefficient for the NCCS was reported as 0.96, indicating strong reliability [36].

Once the participant has completed the pre-intervention questionnaire, participants will receive the Safe Virtual Reality Utilization Guide. This guide is designed to help participants avoid any risks of injury or discomfort while using the HP Reverb G2 headset. After reviewing the guide, the participant will be fitted with the headset and given the paired controllers. Based on their preference, they may be seated or standing, with limited physical movement permitted within a designated, open area.

Each participant will experience approximately 20–30 minutes of IVR simulation. This simulation is divided into two main settings: emergency department (see Figs 1 and 2) and community health centre. They present the narrative of an immigrant woman in crisis, displaying severe anxiety, depressive symptoms, and suicidal ideations. The first scenario requires the participants to assess, stabilize, and establish outpatient community care; while the second scenario involves a more comprehensive assessment of the virtual patient needs, addressing barriers to mental health access, and developing a care plan that harnesses the strengths and resilience of the patient. Throughout this simulation, the participants are expected to communicate and foster therapeutic alliance with the virtual patient by applying culturally appropriate approaches, including cultural competence and humility.

The simulation is designed to be experienced with limited to no guidance required from anyone. Any interactions or choices made by participants in the simulation are not recorded, evaluated, or analyzed, to allow students to freely explore the environment without fear of losing points or formal assessment. When participants select an incorrect dialogue choice, the narrator within the simulation will explain the consequences of that choice and the scene then restarts from that section, giving participants the opportunity to reflect and choose differently the next time.

After the intervention, participants will complete the post-intervention questionnaires. A debriefing session will be held in which the participant will ask about their physical and psychological wellness. The post-intervention questionnaires, which mirror the pre-intervention questionnaires, will be administered following the simulation and debriefing session. Afterward, participants will be thanked for their time and invited to take part in Phase 3. In total, participants will need approximately one hour for Phase 2 data collection.

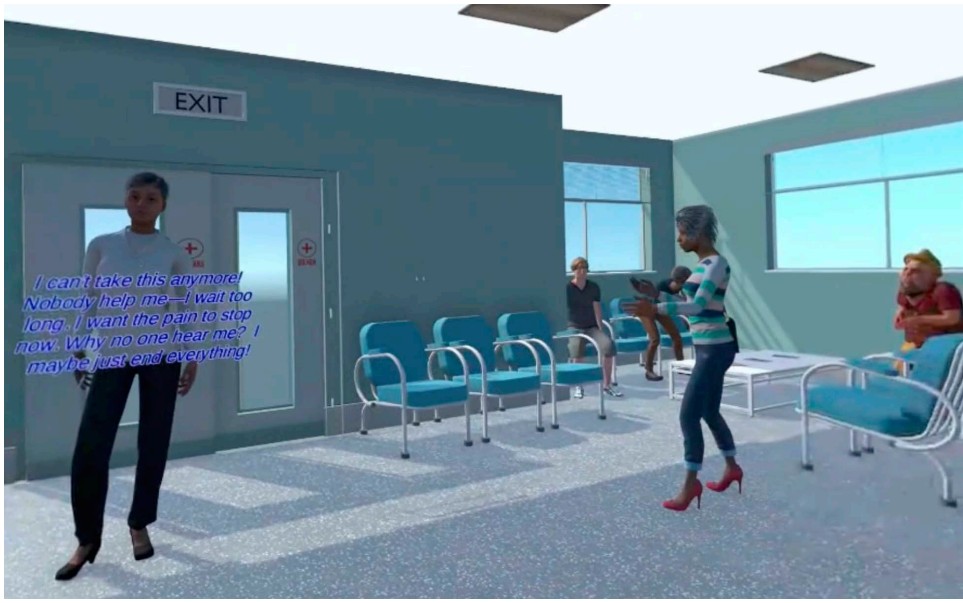

**Fig 1. Virtual patient expressing crisis in the emergency waiting room.**

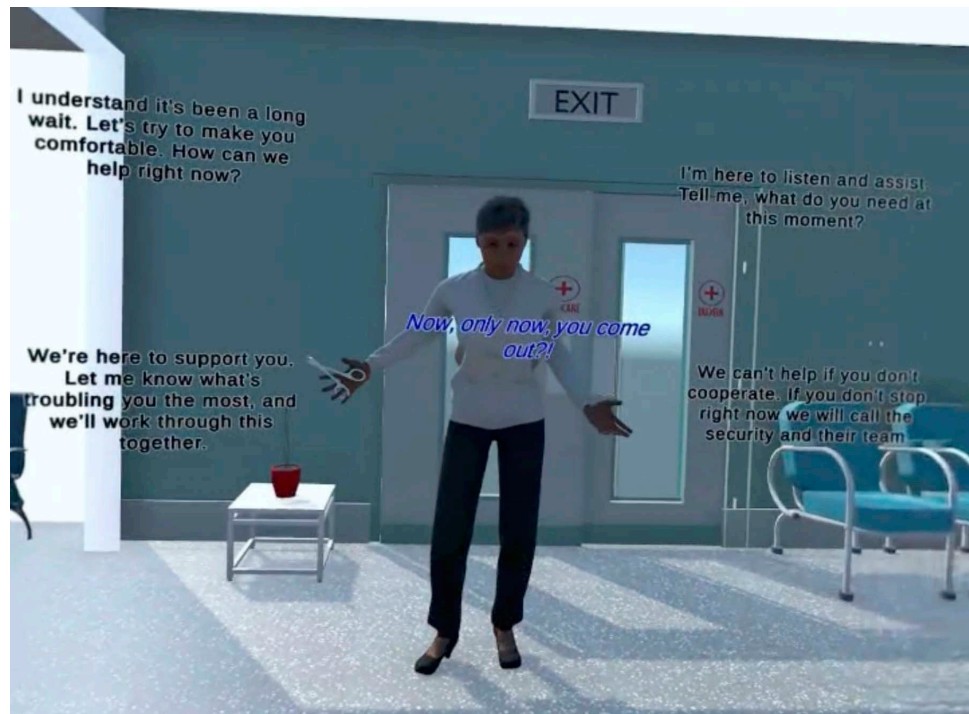

**Fig 2. Dialogue options presented to the participants.**

Once all quantitative data from 60 participants has been collected (i.e., 30 participants from the University of Saskatchewan and 30 from McGill University), data analysis will begin. Descriptive and inferential statistics will be used to analyze the survey data. Initially, a combined analysis of the data collected from the University of Saskatchewan and McGill University will be performed to provide a broad overview of how the simulation performed across different settings. In this combined analysis, paired t-tests or related samples Wilcoxon signed rank tests for each questionnaire scale (i.e., TAMQ, OMS-HC, and NCCS) will be used to compare pre- and post-intervention mean scores and to help determine whether there are significant changes in undergraduate nursing students' technology acceptance, attitudes, and behaviours following the IVR simulation. Moreover, multiple regression analysis will be employed on the combined data to identify predictors of IVR simulation acceptance. Subsequently, site-specific analyses will be conducted. The data from the University of Saskatchewan and McGill University will be analyzed independently, using the same statistical methods as in the combined analysis. This site-specific approach will offer insights into the distinct impacts and acceptability of the IVR simulation at each university. In sum, performing combined and site-specific analyses will not only assist in addressing the research questions, but also provide a thorough understanding of the simulation's preliminary effects and acceptability across diverse contexts, while also capturing the unique characteristics and contextual differences of the two university settings. Findings from this phase will be used to modify the preliminary interview guide and inform Phase 3.

**Phase 3 – Interpretive description approach.** Participants from Phase 2 will be given the opportunity to partake in Phase 3, though their involvement in both phases is not mandatory. Those who agree to participate will engage in a semi-structured interview, lasting between 15–30 minutes, conducted remotely via phone or Zoom that will be audio and/or video recorded and transcribed verbatim. If required, the participant will be invited for a follow-up interview.

This phase will employ concurrent data collection and analysis as well as constant comparative analysis in an iterative manner to ensure "the contextual nature of the data is respected and remains intact" [41]. The interview will follow the preliminary guide, adapted from Aggarwal et al. [42], Karakuş Selçuk and Yanikkerem [43], and Verkuyl et al. [44], that focuses on the acceptability of the simulation and the attitudes and behaviours of the participants towards migrants with mental health difficulties. This guide will be revised according to Phase 2's findings and the iterative process of interpretive description approach. As in the Phase 2 data analysis, combined and site-specific analyses will be conducted for Phase 3 to offer a comprehensive view in addition to nuanced perspectives.

Once quantitative and qualitative data have been analyzed separately, they will then be brought together for an overall interpretation to draw connected conclusions about the findings. Consistent with an explanatory sequential mixed methods design [45], this integration will focus on how qualitative results explain or expand specific quantitative findings by identifying which quantitative trends require further explanation and using qualitative data to provide contextual understanding. For example, if quantitative results indicate patterns or differences in technology acceptance as well as mental health attitudes or behaviours towards migrant populations, the qualitative phase will help uncover underlying reasons, experiences, or contextual factors behind those patterns. This explanatory approach ensures that the qualitative insights are not interpreted in isolation but are directly linked to specific quantitative findings, allowing for a cohesive interpretation. Through this integrated analysis, the study will produce nuanced conclusions that connect statistical trends with participants' perspectives, supporting the development of evidence-based recommendations for practice, education, and research.

## Status and timeline

At the time of this manuscript submission, the study is in Phase 2, with participant recruitment and data collection actively underway. Ethics approvals were obtained from University of Saskatchewan on April 24, 2024 (Application ID: 4726), and for McGill University on May 31, 2024 (Study Number: A05-B31-24B [24-04-089]). Participant recruitment began in September 2024 and is scheduled to conclude within two weeks (by end of March 2025). Phase 2 data collection commenced

in January 2025 and will be completed concurrently with participant recruitment. Following the conclusion of recruitment and data collection, quantitative data analysis will begin immediately and is anticipated to take approximately two weeks.

Phase 3 will commence thereafter and is scheduled for completion within the following two and a half months. Participants from Phase 2 have already been solicited for their participation in Phase 3, with some having already scheduled their semi-structured interviews for April 2025. Upon completion of Phase 3, the writing of the final manuscript will proceed, with plans to submit for publication and present findings at relevant conferences. Study results are expected to be ready for dissemination by the end of July 2025, not accounting for journal review timelines.

## Discussion

This study leverages the power of VR to enhance nursing education and improve mental health care for migrants. The expected outcomes include improved acceptability of VR as a training tool, positive changes in students' attitudes and behaviours, and enhanced cultural competence. This innovative approach has the potential to set a new standard in nursing education, equipping future nurses with the skills and empathy needed to provide high-quality care to all patients, regardless of their background.

Given the rapid globalization, it is crucial for nursing students to be well-prepared to address the varied backgrounds and health needs of individuals, guided by the principles of holistic care, equity, diversity, and inclusion. This project offers an interactive, judgment-free learning environment that extends beyond conventional classroom settings.

Moreover, by focusing on migrants with mental health issues, the study aims to heighten students' understanding of the complex vulnerabilities affecting these groups' access to quality mental health services. This study could lead to improved care delivery and increased interest in mental health careers. Furthermore, this study will involve migrants and nursing students in developing IVR simulation—a novel approach in participatory research. These activities are designed to enhance the affective, empathetic, and communication skills of nursing students through IVR, which remains relatively unexplored [4]. The ultimate goal is to incorporate this simulation into the undergraduate nursing curriculum by enhancing the training and preparedness of future healthcare professionals.

## Author contributions

**Conceptualization:** Geneveave Barbo, Donald Leidl, Pammla Petrucka.

**Methodology:** Geneveave Barbo, Donald Leidl, Marjorie Montreuil, Hua Li, Solina Richter, Pammla Petrucka.

**Software:** Geneveave Barbo.

**Supervision:** Pammla Petrucka.

**Writing – original draft:** Geneveave Barbo.

**Writing – review & editing:** Geneveave Barbo, Donald Leidl, Marjorie Montreuil, Hua Li, Solina Richter, Pammla Petrucka.

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
