## [Decision Letter · Decision Letter 0]

7 Jul 2025

PONE-D-25-14620
Immersive Virtual Reality Simulation for Undergraduate Nursing Students: Enhancing Mental Health Care for Migrants - A Mixed Method Study Protocol
PLOS ONE

Dear Dr. Barbo,

Thank you for submitting your manuscript to PLOS ONE. After careful consideration, we feel that it has merit but does not fully meet PLOS ONE’s publication criteria as it currently stands. Therefore, we invite you to submit a revised version of the manuscript that addresses the points raised during the review process.

We look forward to receiving your revised manuscript.

Kind regards,

Made Satya Nugraha Gautama, RN, M.Sc.,M.N.Sc

Academic Editor

PLOS ONE

Additional Editor Comments:

Dear Authors,

Thank you for submitting your manuscript, "Immersive Virtual Reality Simulation for Undergraduate Nursing Students: Enhancing Mental Health Care for Migrants - A Mixed Method Study Protocol," to Plos One. While we appreciate the innovative approach of utilizing immersive virtual reality (IVR) to address a critical gap in nursing education, particularly in the context of migrant mental health care, we have identified some areas that require further attention before the manuscript can be considered for publication. Specifically, we require greater detail on the development procedure of the IVR intervention. A more thorough description of the design and creation process is needed, to include specific scenarios and content, the theoretical basis for the simulation design, technical aspects, and insights from pilot testing and refinement processes. In addition, while the manuscript mentions that an integrative review will inform the simulation design, it requires a more robust explanation of how these findings translate into specific elements of the IVR experience.

Moreover, we encourage you to consider reviewers points carefully and revise your manuscript accordingly. We look forward to receiving a revised version that addresses these concerns.

Reviewers' comments:

Reviewer's Responses to Questions

**Comments to the Author**

1. Does the manuscript provide a valid rationale for the proposed study, with clearly identified and justified research questions?

Reviewer #1: Yes

Reviewer #2: Yes

2. Is the protocol technically sound and planned in a manner that will lead to a meaningful outcome and allow testing the stated hypotheses?

Reviewer #1: Yes

Reviewer #2: Yes

3. Is the methodology feasible and described in sufficient detail to allow the work to be replicable?

Reviewer #1: Yes

Reviewer #2: Yes

4. Have the authors described where all data underlying the findings will be made available when the study is complete?

Reviewer #1: Yes

Reviewer #2: No

5. Is the manuscript presented in an intelligible fashion and written in standard English?

Reviewer #1: Yes

Reviewer #2: Yes

6. Review Comments to the Author

You may also provide optional suggestions and comments to authors that they might find helpful in planning their study.

Reviewer #1: This study protocol presents a highly relevant and innovative mixed methods study design exploring immersive virtual reality (IVR) simulation to enhance nursing students’ competency in providing mental health care to migrant populations. This topic is of great importance in the global context, and there is a need for an inclusive approach to health care. The stepwise study design suggests that the research methods are sound, however, several important areas need to be clarified and refined before this protocol can be accepted for publication:

1. Ethical Considerations and Participant Protection

The protocol states that ethics approval has been obtained from the University of Saskatchewan and McGill University. As the study is in phase 2, it would be helpful to include the ethics approval number and date of approval. As the participants were nursing students, further clarification is needed regarding how the risk of indirect coercion was avoided, especially if the researcher was their faculty member. Was recruitment conducted by a party not directly involved in the teaching process?

2. Description of Student Participants

a. The description of the students who were included in the study is unclear. Please clarify:

a)Were all cohort years included?

b) Do participants have a specific background related to mental health training?

c) How will racial, cultural, and academic diversity be controlled or analyzed?

b. It is important to explain whether the curricula at the two universities are similar, given that the results obtained will be compared.

3. Simulation Validation and Development Process

It is also necessary to explain how feedback from committee members was collected and analyzed, and how the results were used to revise the simulation.

4. Data Integration in Mixed Methods

It is advisable to explicitly explain the data integration strategy based on mixed methods methodological references, such as Creswell & Plano Clark.

Reviewer #2: The study is highly relevant to the current needs in healthcare education, particularly in preparing nursing students to provide mental health care for migrant populations. However, the protocol lacks sufficient detail regarding Phase One, specifically the development process of the Immersive Virtual Reality (IVR) simulation. It is important to clearly describe how the validated scenarios will be translated into interactive IVR storylines. Additionally, more explanation is needed about the nature of the interaction between participants and the virtual patients—such as whether participants will receive feedback during the simulation and what form that feedback will take. Furthermore, the duration of each IVR session should be specified, as this is critical for understanding the scope and feasibility of the intervention.

7. PLOS authors have the option to publish the peer review history of their article (what does this mean?). If published, this will include your full peer review and any attached files.

Reviewer #1: No

Reviewer #2: **Yes: **Ariani Arista Putri Pertiwi

---

## [Author Response · Author response to Decision Letter 1]

10 Jul 2025

Editor Comments: Dear Authors, Thank you for submitting your manuscript, "Immersive Virtual Reality Simulation for Undergraduate Nursing Students: Enhancing Mental Health Care for Migrants - A Mixed Method Study Protocol," to Plos One. While we appreciate the innovative approach of utilizing immersive virtual reality (IVR) to address a critical gap in nursing education, particularly in the context of migrant mental health care, we have identified some areas that require further attention before the manuscript can be considered for publication. Specifically, we require greater detail on the development procedure of the IVR intervention. A more thorough description of the design and creation process is needed, to include specific scenarios and content, the theoretical basis for the simulation design, technical aspects, and insights from pilot testing and refinement processes.

- Additional information has now been added to the Phase 1 - Development of the Intervention

In addition, while the manuscript mentions that an integrative review will inform the simulation design, it requires a more robust explanation of how these findings translate into specific elements of the IVR experience.

- Additional information has now been added to the Phase 1 - Development of the Intervention

Moreover, we encourage you to consider reviewers points carefully and revise your manuscript accordingly. We look forward to receiving a revised version that addresses these concerns.

- Thank you!

Reviewers' comments:

Have the authors described where all data underlying the findings will be made available when the study is complete? The PLOS Data policy requires authors to make all data underlying the findings described in their manuscript fully available without restriction, with rare exception, at the time of publication. The data should be provided as part of the manuscript or its supporting information, or deposited to a public repository. For example, in addition to summary statistics, the data points behind means, medians and variance measures should be available. If there are restrictions on publicly sharing data—e.g. participant privacy or use of data from a third party—those must be specified. Reviewer #2: No

- This concern has been addressed and clarified in the Ethical Considerations section

Reviewer #1: Ethical Considerations and Participant Protection: The protocol states that ethics approval has been obtained from the University of Saskatchewan and McGill University. As the study is in phase 2, it would be helpful to include the ethics approval number and date of approval.

- Ethics approval numbers and dates of approval has been specified at the Status and Timeline section

As the participants were nursing students, further clarification is needed regarding how the risk of indirect coercion was avoided, especially if the researcher was their faculty member. Was recruitment conducted by a party not directly involved in the teaching process?

- Explanation has been added to the Ethical Considerations section

Description of Student Participants: a. The description of the students who were included in the study is unclear. Please clarify: a)Were all cohort years included? b) Do participants have a specific background related to mental health training? c) How will racial, cultural, and academic diversity be controlled or analyzed? b. It is important to explain whether the curricula at the two universities are similar, given that the results obtained will be compared.

- Explanation has been added to the Ethical Considerations section

Simulation Validation and Development Process. It is also necessary to explain how feedback from committee members was collected and analyzed, and how the results were used to revise the simulation.

- Clarification has now been provided in Phase 1 - Development of the Intervention

Data Integration in Mixed Methods. It is advisable to explicitly explain the data integration strategy based on mixed methods methodological references, such as Creswell & Plano Clark.

- Clarification has now been provided in Phase 3 - Interpretive Description Approach

Reviewer #2: The study is highly relevant to the current needs in healthcare education, particularly in preparing nursing students to provide mental health care for migrant populations. However, the protocol lacks sufficient detail regarding Phase One, specifically the development process of the Immersive Virtual Reality (IVR) simulation. It is important to clearly describe how the validated scenarios will be translated into interactive IVR storylines.

- Additional information has now been added to the Phase 1 - Development of the Intervention

Additionally, more explanation is needed about the nature of the interaction between participants and the virtual patients—such as whether participants will receive feedback during the simulation and what form that feedback will take.

- This is now included in the Phase 2 - One-group Pre- and Post-quasi-experimental section

Furthermore, the duration of each IVR session should be specified, as this is critical for understanding the scope and feasibility of the intervention.

- This is now included in the Phase 2 - One-group Pre- and Post-quasi-experimental

---

## [Editor Report · Decision Letter 1]

6 Aug 2025

PONE-D-25-14620R1
Immersive Virtual Reality Simulation for Undergraduate Nursing Students: Enhancing Mental Health Care for Migrants - A Mixed Method Study Protocol
PLOS ONE

Dear Dr. Barbo,

Thank you for submitting your manuscript to PLOS ONE. After careful consideration, we feel that it has merit but does not fully meet PLOS ONE’s publication criteria as it currently stands. Therefore, we invite you to submit a revised version of the manuscript that addresses the points raised during the review process.

We look forward to receiving your revised manuscript.

Kind regards,

Made Satya Nugraha Gautama, RN, M.Sc.,M.N.Sc

Academic Editor

PLOS ONE

Journal Requirements:

Additional Editor Comments:

Dear author(s), thank you for updated manuscript. Please explain specifically and technically how the Immersive Virtual Reality Simulation are delivered, such as:

- VR device used (what specific VR hardware use, how much the resolution, are there inside-out tracking and wireless mobility?)

- technical procedure (Development Phase; Hardware Setup; and Simulation Run)

- Provide the Content of the Simulation (visual; screen capture as supplement file/include within the manuscript)

Reviewers' comments:

None

---

## [Author Response · Author response to Decision Letter 2]

7 Aug 2025

Dear author(s), thank you for updated manuscript. Please explain specifically and technically how the Immersive Virtual Reality Simulation are delivered, such as:

- VR device used (what specific VR hardware use, how much the resolution, are there inside-out tracking and wireless mobility?)

o This information has now been included under Phase 2 - One-group Pre- and Post-quasi-experimental

- Technical procedure (Development Phase; Hardware Setup; and Simulation Run)

o This information has now been included under Phase 1 - Development of the Intervention and Phase 2 - One-group Pre- and Post-quasi-experimental

- Provide the Content of the Simulation (visual; screen capture as supplement file/include within the manuscript)

o Figures 1 and 2 have now been uploaded and included.

---

## [Editor Report · Decision Letter 2]

1 Sep 2025

PONE-D-25-14620R2
Immersive Virtual Reality Simulation for Undergraduate Nursing Students: Enhancing Mental Health Care for Migrants - A Mixed Method Study Protocol
PLOS ONE

Dear Dr. Barbo,

Thank you for submitting your manuscript to PLOS ONE. After careful consideration, we feel that it has merit but does not fully meet PLOS ONE’s publication criteria as it currently stands. Therefore, we invite you to submit a revised version of the manuscript that addresses the points raised during the review process.

Dear author(s), thank you for updated manuscript. Please explain specifically and technically how the Immersive Virtual Reality Simulation are delivered, such as:

- VR device used (what specific VR hardware use, how much the resolution, are there inside-out tracking and wireless mobility?)

- technical procedure (Development Phase; Hardware Setup; and Simulation Run)

- Provide the Content of the Simulation (visual; screen capture as supplement file/include within the manuscript)

We look forward to receiving your revised manuscript.

Kind regards,

Made Satya Nugraha Gautama, RN, M.Sc.,M.N.Sc

Academic Editor

PLOS ONE
---

## [Author Response · Author response to Decision Letter 3]

11 Sep 2025

Dear author(s), thank you for updated manuscript. Please explain specifically and technically how the Immersive Virtual Reality Simulation are delivered, such as:

- VR device used (what specific VR hardware use, how much the resolution, are there inside-out tracking and wireless mobility?)

o This information has now been included under Phase 2 - One-group Pre- and Post-quasi-experimental

- Technical procedure (Development Phase; Hardware Setup; and Simulation Run)

o This information has now been included under Phase 1 - Development of the Intervention and Phase 2 - One-group Pre- and Post-quasi-experimental

- Provide the Content of the Simulation (visual; screen capture as supplement file/include within the manuscript)

o Figures 1 and 2 have now been uploaded and included.

---

## [Editor Report · Decision Letter 3]

15 Sep 2025

Immersive Virtual Reality Simulation for Undergraduate Nursing Students: Enhancing Mental Health Care for Migrants - A Mixed Method Study Protocol

PONE-D-25-14620R3

Dear Dr. Barbo,

We’re pleased to inform you that your manuscript has been judged scientifically suitable for publication and will be formally accepted for publication once it meets all outstanding technical requirements.

Kind regards,

Made Satya Nugraha Gautama, RN, M.Sc.,M.N.Sc

Academic Editor

PLOS ONE
---

## [Editor Report · Acceptance letter]

PONE-D-25-14620R3

PLOS ONE

Dear Dr. Barbo,

I'm pleased to inform you that your manuscript has been deemed suitable for publication in PLOS ONE. Congratulations! Your manuscript is now being handed over to our production team.

Kind regards,

on behalf of

Mr. Made Satya Nugraha Gautama

Academic Editor

PLOS ONE